# Host chitinase 3-like-1 is a universal therapeutic target for SARS-CoV-2 viral variants in COVID-19

**Suchitra Kamle[1], Bing Ma[1], Chang Min Lee[1], Gail Schor[1], Yang Zhou[1], Chun Geun Lee[1], Jack A Elias[2]\***

[1]Department of Molecular Microbiology and Immunology, Brown University, Providence, United States; [2]Brown University, Providence, United States

**Abstract** Coronavirus disease 2019 (COVID-19) is the disease caused by severe acute respiratory syndrome coronavirus-2 (SARS-CoV-2; SC2), which has caused a worldwide pandemic with striking morbidity and mortality. Evaluation of SC2 strains demonstrated impressive genetic variability, and many of these viral variants are now defined as variants of concern (VOC) that cause enhanced transmissibility, decreased susceptibility to antibody neutralization or therapeutics, and/or the ability to induce severe disease. Currently, the delta (δ) and omicron ( o ) variants are particularly problematic based on their impressive and unprecedented transmissibility and ability to cause breakthrough infections. The delta variant also accumulates at high concentrations in host tissues and has caused waves of lethal disease. Because studies from our laboratory have demonstrated that chitinase 3-like-1 (CHI3L1) stimulates ACE2 and Spike (S) priming proteases that mediate SC2 infection, studies were undertaken to determine if interventions that target CHI3L1 are effective inhibitors of SC2 viral variant infection. Here, we demonstrate that CHI3L1 augments epithelial cell infection by pseudoviruses that express the alpha, beta, gamma, delta, or omicron S proteins and that the CHI3L1 inhibitors anti-CHI3L1 and kasugamycin inhibit epithelial cell infection by these VOC pseudovirus moieties. Thus, CHI3L1 is a universal, VOC-independent therapeutic target in COVID-19.

**\*For correspondence:**
jack_elias@brown.edu

## Editor's evaluation

In this article, Kamle and colleagues report that inhibition of host constitutively expressed chitinase 3-like-1 (CHI3L1) increased epithelial expression of ACE2 and SPP, resulting in epithelial cell viral uptake of pseudoviruses that express the α, β, γ, δ, or omicron S proteins, and they further show that antagonism of CHI3L1 using anti-CHI3L1 or kasugamycin inhibits epithelial cell infection by the pseudoviruses with ancestral, α, β, γ S protein mutations. The in vitro data have relevance to SARS-CoV-2 pathogenesis and potentially has therapeutic implications in that the anti-CHI3L1 antibody and/or kasugamycin might be a treatment for this pandemic virus. These in vitro data are novel, and the results are clear and convincing. The authors acknowledge the limitation of the lack of in vivo data, and the hope is that the publication of this study will encourage a collaboration where those data can be obtained.

## Introduction

Coronavirus disease 2019 (COVID-19), the illness caused by severe acute respiratory syndrome coronavirus virus-2 (SARS-CoV-2; SC2), was first discovered in man in 2019 and declared a global pandemic by the World Health Organization on March 11, 2020. It is the cause of a global health crisis with

countries experiencing multiple waves of illness, resulting in more than 273 million confirmed clinical cases and more than 5.34 million deaths as of December 17, 2021 (*Chen et al., 2020*; *Huang et al., 2020*; *Li et al., 2020*; *Wang et al., 2020*; *Guan et al., 2020*; *Wu and McGoogan, 2020*; *CSSE JHU, 2021*; *Aleem et al., 2021*). The disease caused by SC2 was initially noted to manifest as a pneumonia (*Lake, 2020*). It is now known to have impressive extrapulmonary manifestations and vary in severity from asymptomatic to mildly symptomatic to severe disease with organ failure to death (*Inciardi et al., 2020*; *Gupta et al., 2020*). However, the cellular and molecular events that account for the multiple waves of disease and the impressive clinical and pathological heterogeneity that have been seen have not been defined.

SC2 interacts with cells via its Spike (S) protein, which binds to its cellular receptor angiotensin-converting enzyme 2 (ACE2) (*Millet and Whittaker, 2015*; *Coutard et al., 2020*; *Hoffmann et al., 2020*; *Letko et al., 2020*). To mediate viral entry, the S protein is processed into S1 and S2 subunits by the S priming proteases (SPP), including TMPRSS2, cathepsin L (CTSL), and, to a lesser degree, FURIN. The S2 subunit mediates the fusion of the viral envelope and cell plasma membrane to allow for virus–cell entry (*Hoffmann et al., 2020*; *Bourgonje et al., 2020*; *Zhang et al., 2020*). In keeping with the importance of virus–cell interactions, many treatments for COVID-19 have focused on disease prevention using nonpharmacological public health measures, antiviral antibodies, and a critical global vaccination strategy (*Aleem et al., 2021*). Treatments of acute infection include supportive interventions, anti-inflammatories, recently described oral antivirals, and direct antivirals such as remdesivir (*Aleem et al., 2021*). Surprisingly, although ACE2 and SPP play critical roles in SC2 infection and proliferation, therapeutics that focus on these host moieties have not been adequately investigated.

Early strains of SC2 from Wuhan, China, manifest limited genetic diversity (*Baric, 2020*). However, genetic epidemiological evidence in February 2020 demonstrated the global emergence of a new dominant SC2 variant called D614G (*Baric, 2020*; *Korber et al., 2020*). This variant was associated with enhanced transmissibility based on an S protein that is more likely to assume an 'open' configuration and bind ACE2 with enhanced avidity when compared to the ancestral strain (*Aleem et al., 2021*; *Baric, 2020*). In the interval, since then multiple other SC2 variants have been appreciated. Many are now defined as variants of concern (VOC) due to their enhanced transmissibility, decreased susceptibility to neutralization by antibodies obtained from natural infection or vaccination, ability to evade detection, or ability to decrease therapeutic or vaccine effectiveness (*Aleem et al., 2021*). As of June 2021, four variants (alpha, beta, gamma, and delta) had been defined as VOC. Most recently, omicron has been added to the list of VOC. Of these viral moieties, delta and omicron (B.1.1.529) are most problematic. The delta variant has caused the deadly second wave of disease in India and waves of COVID-19 at other sites around the world (*Del Rio et al., 2021*; *Dhar et al., 2021*). It is also known to have a high level of transmissibility and virulence when compared to ancestral controls and the alpha (α), beta (β), and gamma (g) variants. It manifests an enhanced ability to replicate and accumulates at very high levels in airways and tissues (*Aleem et al., 2021*). In keeping with these characteristics, recent studies have also demonstrated that delta is associated with breakthrough infections in vaccinated individuals and a decrease in vaccine effectiveness, especially in the elderly (*Shastri et al., 2021*). The omicron variant was first detected in November 2021 and quickly declared a VOC based on its impressive transmissibility (*Cameroni et al., 2022*). It has 37 S protein mutations in its predominant haplotype, 15 of which are in its receptor-binding domain (RBD), which is the major target of neutralizing antibodies (*Cameroni et al., 2022*; *Sheikh et al., 2021*; *Greaney et al., 2021*). Although it appears to cause less severe disease than delta, its impressive ability to spread and resist antibody neutralization has resulted in surges that run the risk of overwhelming health-care systems worldwide. In spite of the important differences in the S proteins of the variants and the impressive importance of S and ACE2 in COVID-19, therapies that focus on host targets such as CHI3L1, ACE2, and SPP that are effective in multiple SC2 variants have not been adequately defined.

We hypothesized that therapies that target the host factors involved in SC2 infection like CHI3L1 can contribute to the control of COVID-19 induced by all viral variants that use ACE2. To test this hypothesis, we employed pseudoviruses that expressed S proteins from the α, β, γ, δ, and o variants and assessed the ability of CHI3L1-based interventions to modify their ability to infect human lung epithelial cells. These studies demonstrate that CHI3L1 augments the expression and accumulation of ACE2 and SPP and augments epithelial infection by the α, β, γ, δ, and o pseudovirus variants. They also demonstrate that anti-CHI3L1 and the small-molecule CHI3L1 inhibitor kasugamycin both inhibit

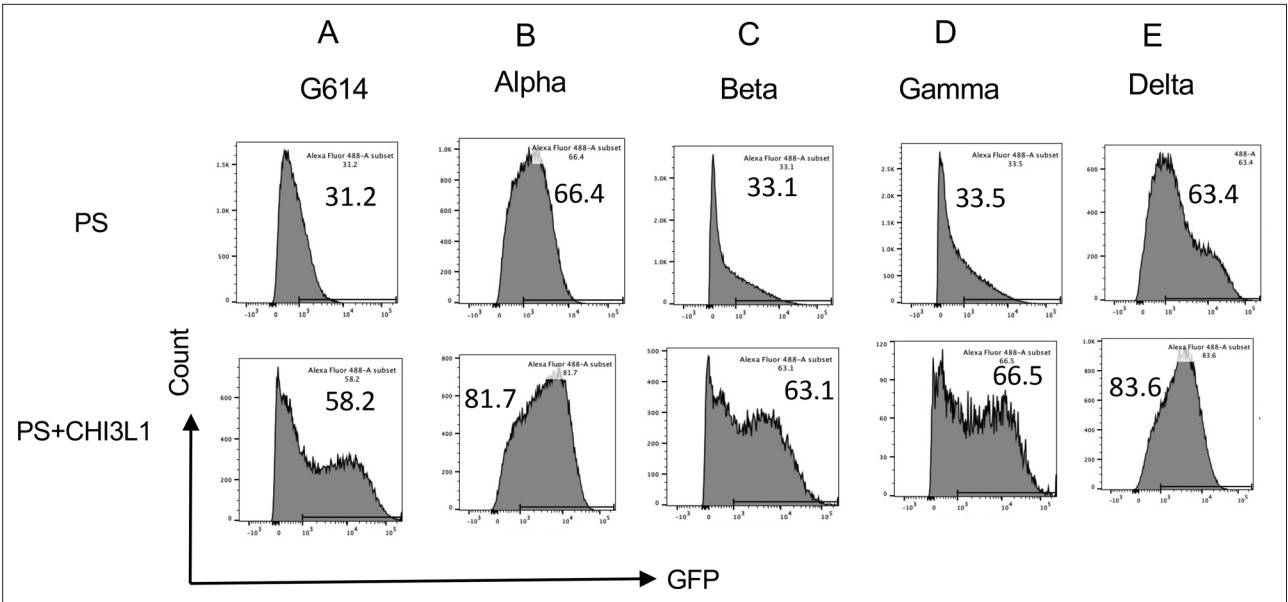

**Figure 1.** CHI3L1 stimulation of pseudovirus uptake. Calu-3 cells were incubated with recombinant human (rh) CHI3L1 (CHI3L1, 250 ng/ml) or vehicle (PBS) control for 48 hr. Pseudoviruses (PS) that contain S proteins with the (**A**) G614, (**B**) alpha, (**C**) beta, (**D**) gamma, or (**E**) delta mutations were added, and GFP was quantitated by fluorescence-activated cell sorting (FACS). The percentage of GFP-positive cells was evaluated by flow cytometry. The noted values are representative of a minimum of three similar evaluations.

the expression and accumulation of epithelial ACE2 and SPP and, in turn, inhibit epithelial infection by pseudoviruses that contain the α, β, γ, δ, and ο S proteins.

## Results

### CHI3L1 stimulates epithelial cell uptake of the G614 and the alpha, beta, and gamma pseudotyped SC2 viruses

Previous studies from our laboratory demonstrated that CHI3L1 is a potent stimulator of epithelial expression of ACE2 and SPP and epithelial cell viral uptake (*Kamle et al., 2021*). To determine if the major S variants altered these responses, we compared the uptake of pseudovirus with ancestral and mutated S proteins by untreated and CHI3L1-treated Calu-3 cells. As can be seen in *Figure 1A*, CHI3L1 was a potent stimulator of the uptake of pseudovirus with the ancestral G614 S protein. Similar increases in Calu-3 cell pseudovirus uptake were seen when the S proteins that are characteristic of the α, β, or γ variants were employed (*Figure 1B–D*). When viewed in combination, these studies demonstrate that CHI3L1 augments SC2 pseudovirus uptake when the ancestral D614G and α, β, or γ S protein mutations are present.

### CHI3L1 stimulates epithelial cell uptake of delta pseudotyped SC2 viruses

Because the delta SC2 variant manifests enhanced viral infectivity and has spread widely since it first appeared in December 2020, the ability of CHI3L1 to alter its ability to infect human epithelial cells was also assessed. In these experiments, CHI3L1 was also a potent stimulator of the uptake of pseudovirus with delta S protein mutations (*Figure 1E*). The findings noted above and these observations, in combination, demonstrate that CHI3L1 is a stimulator of human epithelial cell uptake of SC2 viral pseudotypes with S protein mutations from the α, β, γ, and δ VOC.

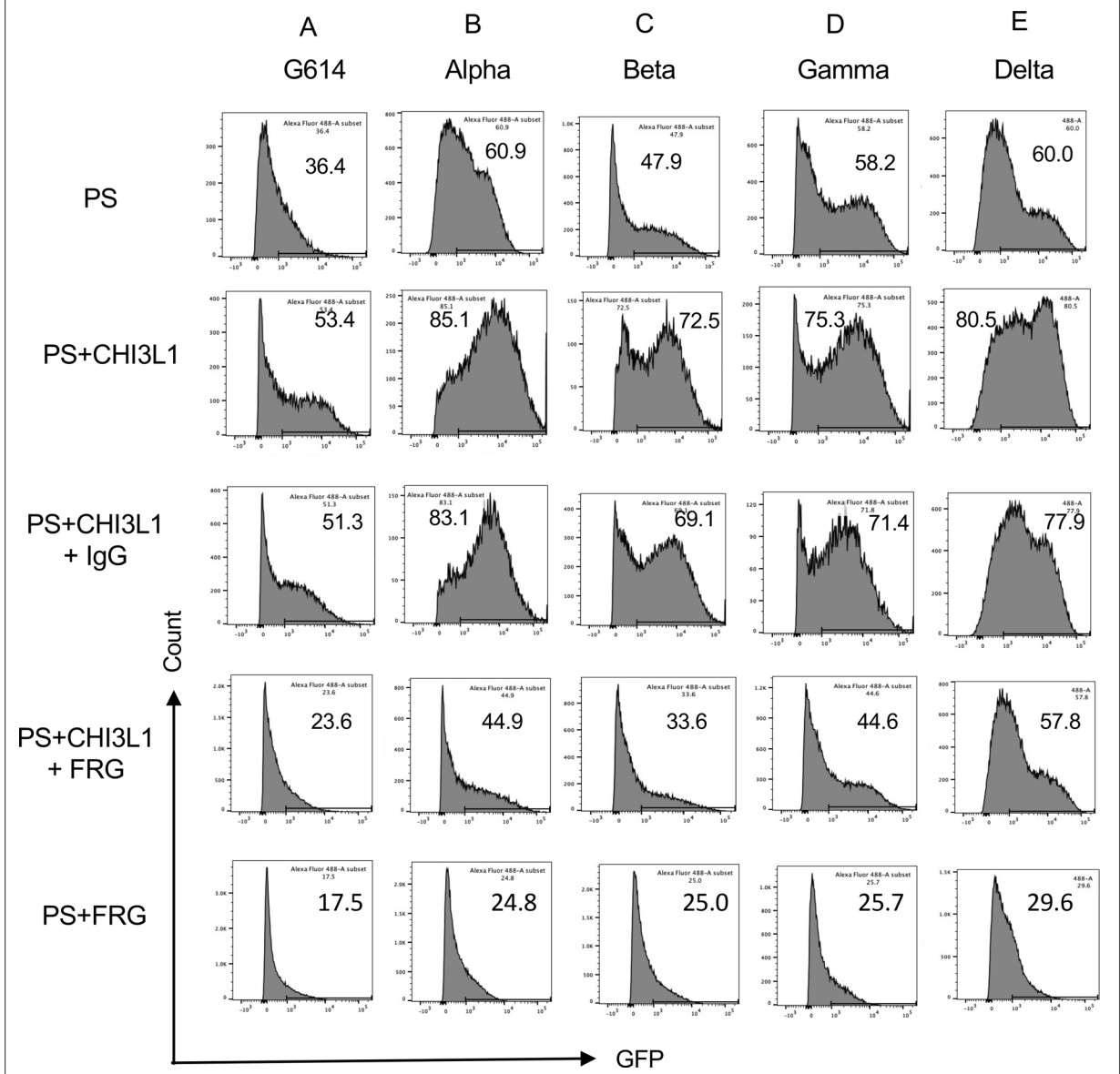

**Figure 2.** Effects of FRG on G614, alpha, beta, gamma, and delta pseudovirus infection. Calu-3 cells were incubated with rhCHI3L1 (CHI3L1, 250 ng/ml) or vehicle control for 48 hr in the presence of anti-CHI3L1 (FRG) or its isotype control (IgG). Pseudoviruses (PS) that contain S proteins with the (**A**) G614, (**B**) alpha, (**C**) beta, (**D**) gamma, or (**E**) delta mutations were added, and GFP was quantitated by fluorescence-activated cell sorting (FACS). The percentage of GFP-positive cells was evaluated by flow cytometry. The noted values are representative of a minimum of three similar evaluations.

### The monoclonal antibody 'FRG' abrogates the CHI3L1-induced increase in epithelial cell uptake of the G614 and the alpha, beta, and gamma pseudotyped viral variants

Studies were next undertaken to define the effects of the monoclonal anti-CHI3L1 antibody entitled 'FRG' on the uptake of pseudovirus by Calu-3 cells treated with and without CHI3L1. As was seen with the ancestral G614 S protein mutation, treatment of Calu-3 cells with rCHI3L1 augmented pseudovirus uptake and FRG abrogated this increase while treatment with the IgG control did not (**Figure 2A**). Interestingly, FRG also diminished pseudovirus uptake by Calu-3 cells even when exogenous rCHI3L1 was not administered (**Figure 2A**). rCHI3L1 had similar stimulatory effects in experiments using pseudovirus with α, β, or γ S protein mutations (**Figure 2B–D**). Importantly, the uptake of pseudoviruses with each of the S mutations in cells treated with and without rCHI3L1 was markedly diminished by FRG as well (**Figure 2A–D**). When viewed in combination, these studies demonstrate

that monoclonal anti-Chi3l1 targeting exogenous and/or endogenous CHI3L1 effectively inhibits the uptake of pseudovirus with ancestral, α, β, or γ S protein mutations.

### The monoclonal antibody 'FRG' abrogates the CHI3L1-induced increase in epithelial cell uptake of the delta pseudotyped viral variants

Because the delta SC2 variant has had such impressive clinical effects, the ability of FRG to alter its ability to infect human epithelial cells was also assessed. Fluorescence-activated cell sorting (FACS)-based evaluations demonstrated that CHI3L1 was a potent stimulator of the uptake of the pseudo-virus with delta S protein mutations (*Figure 2E*). FRG abrogated this increase while treatment with the IgG control did not (*Figure 2E*). Interestingly, FRG also diminished pseudovirus uptake by Calu-3 cells even when exogenous CHI3L1 was not administered (*Figure 2E*). These findings were reinforced by immunocyotchemical evaluations. These studies demonstrated that CHI3L1 augmented Calu-3 cell ACE2 accumulation and delta pseudovirus infection (*Figure 3*). They also demonstrated that FRG abrogated the expression of ACE2 and delta pseudovirus infection at baseline and/or after the administration of rCHI3L1 (*Figure 3*). When viewed in combination, these studies demonstrate that monoclonal anti-CHI3L1 antibody (FRG) targeting exogenous and/or endogenous CHI3L1 effectively inhibits the expression of ACE2 and the uptake of pseudoviruses with the delta or other S protein mutations.

### Kasugamycin is a small molecule with strong anti-CHI3L1 activity

Recent studies from our lab and others identified that Kasugamycin, an aminoglycoside antibiotic, is a novel small molecule that has a strong anti-Chitinase 1 (CHIT1) activities (*Lee et al., 2021*; *Qi et al., 2021*). Since CHIT1 and CHI3L1 share structural homologies as members of 18-glycohydrolase family, we tested whether kasugamycin (KSM) inhibits CHI3L1 activity similarly to CHIT1. As shown in *Figure 4*, KSM treatment abrogated CHI3L1 stimulated ERK and AKT activation in Calu-3 cells, suggesting a strong anti-CHI3L1 activity of KSM.

### Kasugamycin abrogates the CHI3L1-induced increase in epithelial cell uptake of the alpha, beta, and gamma pseudovirus variants

Studies were next undertaken to define the effects of kasugamycin on the uptake of pseudovirus by Calu-3 cells treated with or without rCHI3L1. As seen with the ancestral G614 S protein mutation, treatment of Calu-3 cells with rCHI3L1 augmented ancestral pseudovirus uptake and kasugamycin abrogated these stimulatory effects (*Figure 5A*). rCHI3L1 had similar stimulatory effects in experiments using pseudoviruses with α, β, or γ S protein mutations (*Figure 5B–D*), and these stimulatory effects were markedly decreased by kasugamycin (*Figure 5A–D*). When viewed in combination, these studies demonstrate that kasugamycin targeting exogenous and/or endogenous CHI3L1 effectively inhibits the uptake of pseudovirus with ancestral, α, β, or γ S protein mutations.

### Kasugamycin targeting of CHI3L1 inhibits the uptake of pseudovirus with the delta S protein mutation

Because of the importance of the delta SC2 viral variant, the ability of kasugamycin to alter the variant's ability to infect human epithelial cells was also assessed. CHI3L1 was a potent stimulator of the uptake of pseudoviruses that contain the delta S protein mutations (*Figure 5E*). Kasugamycin abrogated this increase while treatment with the vehicle control did not (*Figure 5E*). Kasugamycin also diminished pseudovirus uptake by Calu-3 cells even when exogenous CHI3L1 was not administered (*Figure 5E*). These findings were reinforced by immunocytochemical evaluations. These studies demonstrated that CHI3L1 augmented Calu-3 cell ACE2 accumulation and delta pseudovirus infection (*Figure 6*). They also demonstrated that FRG abrogated the expression of delta pseudovirus infection at baseline and/or after the administration of rCHI3L1 (*Figure 6*). When viewed in combination, these studies demonstrate that kasugamycin targeting exogenous and/or endogenous CHI3L1 effectively inhibits the uptake of pseudovirus with the alpha, beta, gamma, or delta S protein mutations.

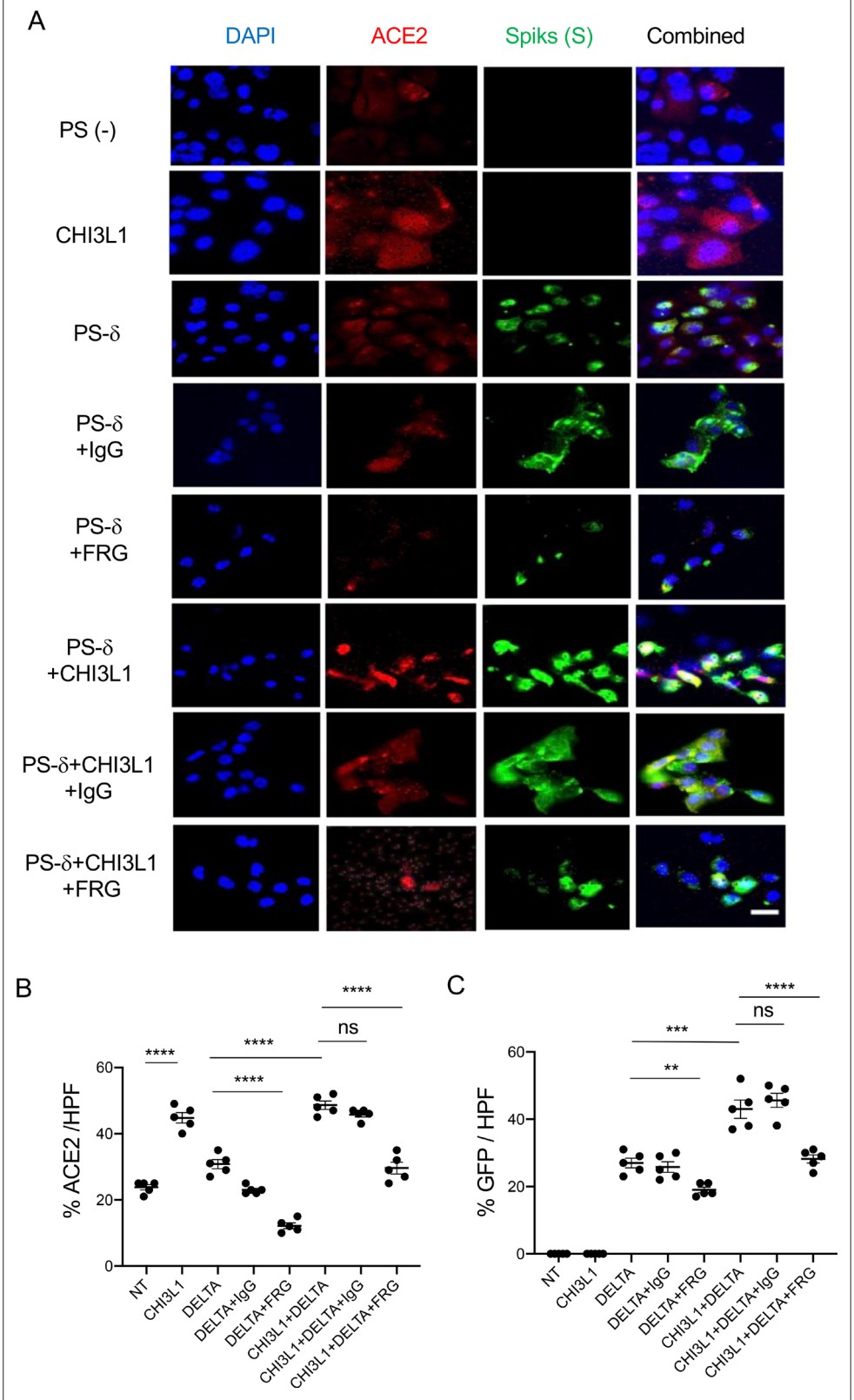

**Figure 3.** Immunocytochemical evaluation of delta pseudovirus infection of Calu-3 cells. Calu-3 cells were incubated in the presence and/or absence of rhCHI3L1 (CHI3L1) in the presence of FRG or its isotype control. (**A**) Pseudoviruses with delta S proteins (PS-δ) were added, and ACE2 and GFP viral infection were evaluated using double-labeled immunocytochemistry (ICC). DAPI (blue) was used to evaluate nuclei, red label was used to

*Figure 3 continued on next page*

*Figure 3 continued*

evaluate ACE2, and the pseudoviruses contained GFP. (**B, C**) The quantification of ACE2 can be seen in panel (**B**), and the quantification of GFP is illustrated in panel (**C**). These evaluations were done using fluorescent microscopy (×20 of original magnification). In these quantifications, five randomly selected fields were evaluated. The values in panels (**B, C**) are the mean ± SEM of the noted five evaluations. **p<0.01; ***p<0.001, ****p<0.0001; ns, not significant (one-way ANOVA with multiple comparisons). Scale bar:10 μm (applies to all subpanels in **A**).

The online version of this article includes the following source data for figure 3:

**Source data 1.** Composite images of immunocytochemical evaluation of delta pseudovirus infection of Calu-3 cells.

## The monoclonal antibody 'FRG' and kasugamycin inhibit epithelial uptake of omicron pseudotyped viral variants

Omicron variant has rapidly spread from its first appreciation as a highly mutated variant causing a localized outbreak in South Africa to the most common SC2 variant in the United States and the world (https://covid.cdc.gov/covid-data-tracker/#variant-proportions). Thus, studies were undertaken to determine if the therapies described above that target CHI3L1, ACE2, and SPP in ancestral, alpha, beta, gamma, and delta pseudoviruses are also effective in pseudoviruses with omicron S protein mutations. As was seen with the α, β, γ, and δ pseudoviruses, CHI3L1 was a potent stimulator of

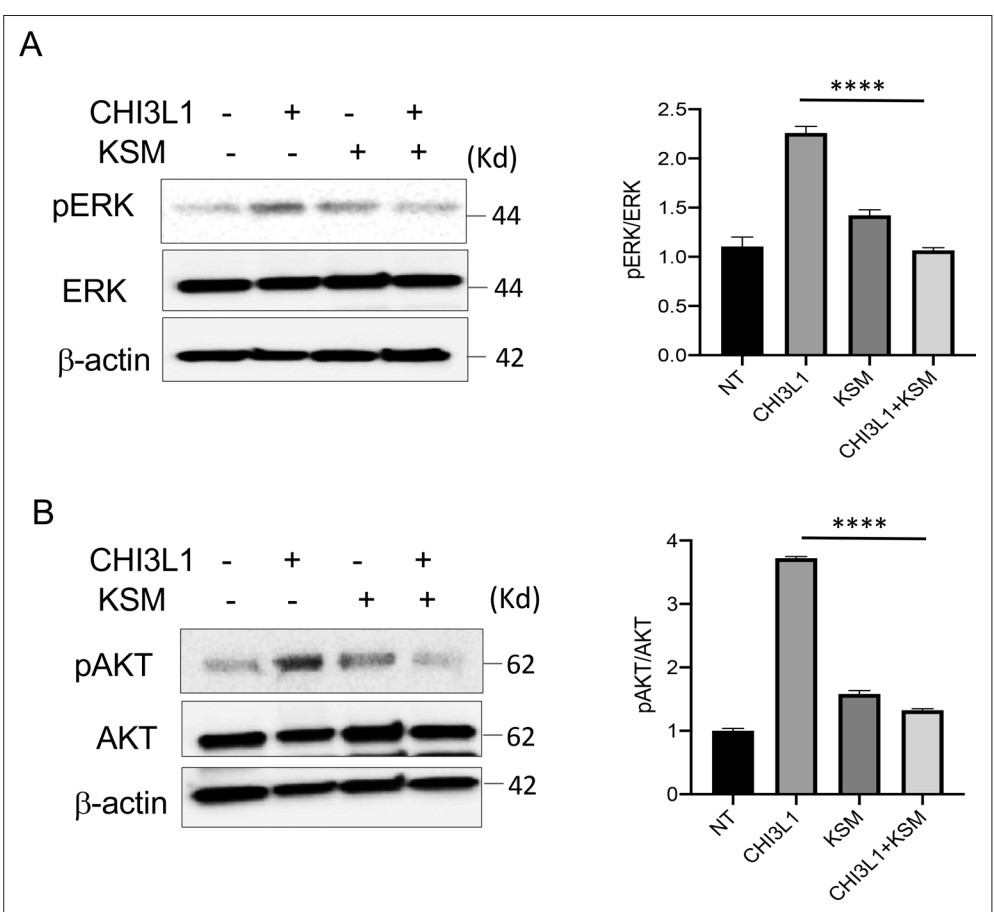

**Figure 4.** Kasugamycin inhibition of CHI3L1-induced signaling. Calu-3 cells were stimulated with rhCHI3L1 (250 ng/ml) or its vehicle control for 2 hr in the presence of kasugamycin (250 ng/ml) and vehicle control (PBS). (**A, B**) Western blotting and densitometry analysis were then employed to evaluate the levels of activated (phosphorylated) (p) and total ERK (**A**) and AKT (**B**). The noted figure is representative of a minimum of three similar experiments. ***p<0.0001 (*t*-test).

The online version of this article includes the following source data for figure 4:

**Source data 1.** Uncut full gel photo for Western blots used in *Figure 4*.

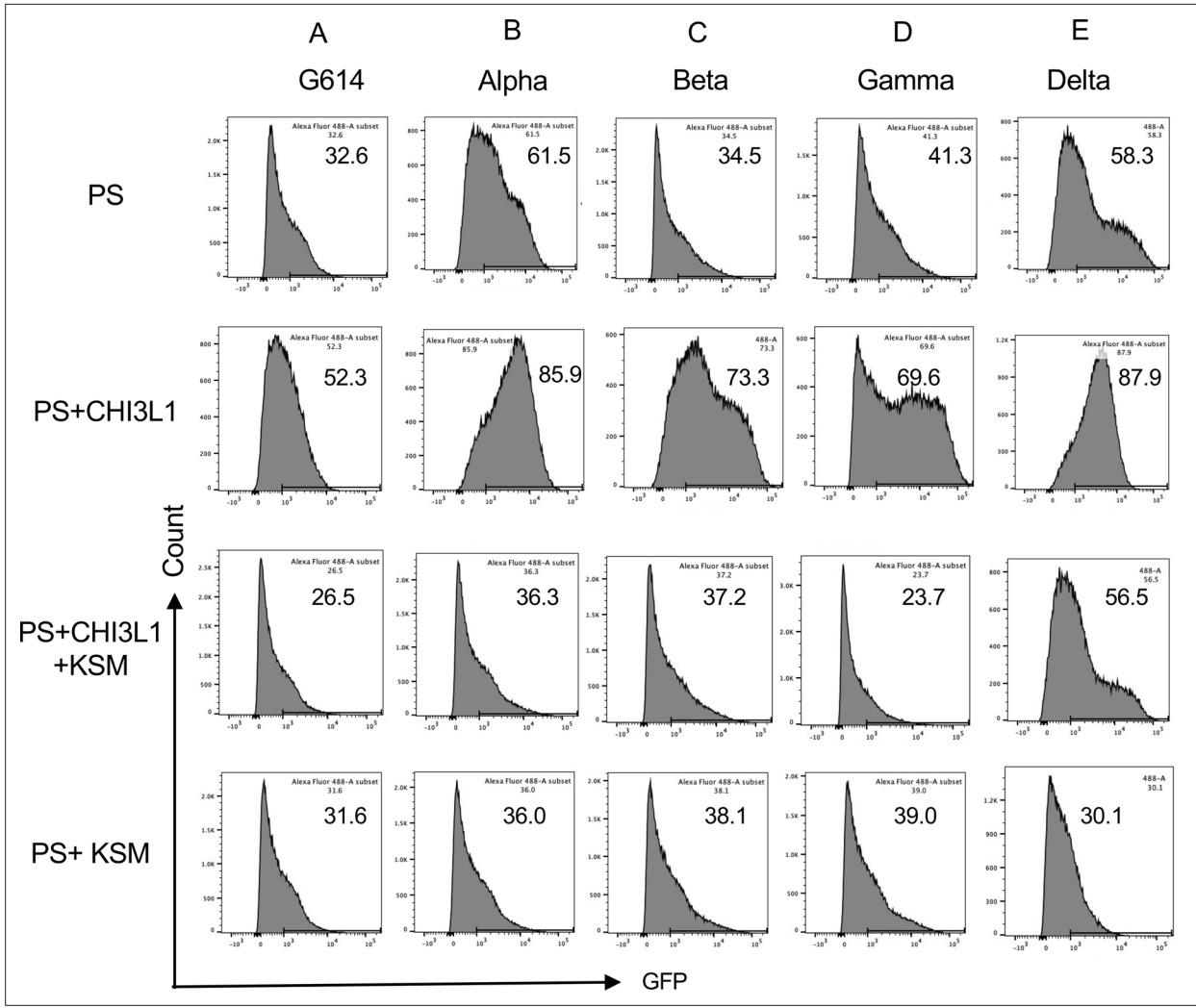

**Figure 5.** Effects of kasugamycin on alpha, beta, gamma, and delta pseudovirus infection. Calu-3 cells were incubated with rhCHI3L1 (250 ng/ml) or vehicle control for 48 hr in the presence of kasugamycin or its vehicle control. Pseudoviruses that contain S proteins with the (**A**) G614, (**B**) alpha, (**C**) beta, (**D**) gamma, or (**E**) delta mutations were added, and GFP was quantitated by fluorescence-activated cell sorting (FACS) analysis. The percentage of GFP-positive cells was evaluated by flow cytometry. The noted values are representative of a minimum of three similar evaluations.

the uptake of pseudoviruses that contained omicron S proteins (*Figure 7*). In addition, FRG and kasugamycin both inhibited epithelial cell uptake of pseudoviruses with omicron S protein mutations (*Figure 7*). These studies demonstrate that antibodies or small-molecule inhibitors that target CHI3L1 inhibit epithelial uptake of pseudoviruses with a wide range of S protein mutations, including those seen in the omicron SC2 variant.

## Discussion

Coronaviruses are large enveloped single-stranded viruses (*Aleem et al., 2021*). Generally, the rates of nucleotide substitution of RNA viruses are fast and mainly the result of natural selection (*Giovanetti et al., 2021*). This high error rate and the subsequent rapidly evolving virus populations can lead to the accumulation of amino acid mutations that affect the transmissibility, cell tropism, pathogenicity, and/or the responsiveness to vaccinations and/or therapies (*Aleem et al., 2021*; *Giovanetti et al., 2021*). The SC2 VOC are known to manifest enhanced transmissibility and diminished vaccine effectiveness when compared to ancestral controls (*Dubey et al., 2021*; *Chia et al., 2021*). Their mutations are important causes of viral infection, the cause of new waves of illness and death and drivers of pandemic persistence (*Dubey et al., 2021*). This can be readily appreciated in the rapid spread

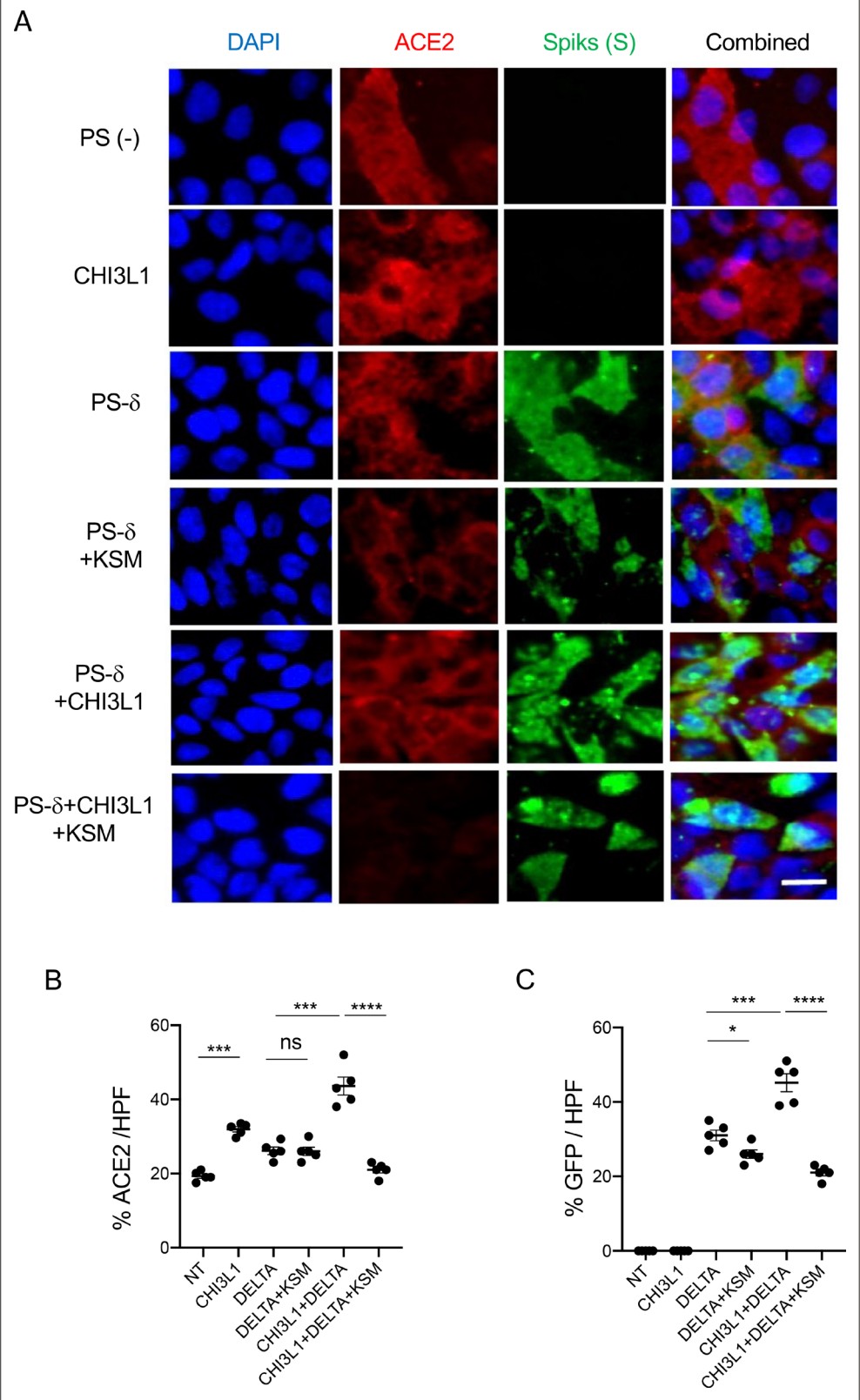

**Figure 6.** Immunocytotochemical evaluation of delta pseudovirus infection of Calu-3 cells. Calu-3 cells were incubated in the presence and/or absence of CHI3L1 (250 ng/ml) in the presence or of kasugamycin (250 ng/ml) or its vehicle control. (**A**) Pseudoviruses with delta S proteins were added, and ACE2 and GFP viral infection were evaluated using double-labeled immunocytochemistry (ICC). DAPI (blue) was used to evaluate nuclei, red label

*Figure 6 continued on next page*

*Figure 6 continued*

was used to evaluate ACE2, and the pseudoviruses contained GFP. (**B, C**) The quantification of ACE2 can be seen in panel (**B**), and the quantification of GFP is illustrated in panel (**C**). These evaluations were done using fluorescent microscopy (×20 of original magnification). In these quantifications, five randomly selected fields were evaluated. The values in panels (**B, C**) are the mean ± SEM of the noted five evaluations. *p<0.05, **p<0.01; ***p<0.001; ns, not significant (one-way ANOVA with multiple comparisons). Scale bar:10 µm (applies to all subpanels in **A**).

The online version of this article includes the following source data for figure 6:

**Source data 1.** Composite images of immunocytotochemical evaluation of delta pseudovirus infection of Calu-3 cells.

of the delta and omicron variants with the latter now accounting for 99.7% of SC2 infections in the United States (https://covid.cdc.gov/covid-data-tracker/#variant-proportions, as of May 14, 2022). It can also be seen in delta's enhanced ability to replicate that drives the viral load up beyond what many other variants can do and outpaces the body's initial antiviral response (*Mlcochova et al., 2021*). The fact that the spectrum of mutations in and characteristics of these variants differs from one another has complicated approaches to vaccination and therapy. In light of the importance of the variants, especially delta and omicron, in COVID-19 studies were undertaken to determine if therapies could be developed by targeting host moieties that help to control many of the major VOC of SC2. In keeping with the importance of ACE2 and SPP in SC2 infection and the impressive ability of CHI3L1 to stimulate these moieties, these studies focused on the relationships between CHI3L1 and ACE2 in infections caused by the α, β γ, δ, and ο variants. They demonstrate that CHI3L1 stimulates the infection caused by these VOC by stimulating the expression and accumulation of ACE2 and SPP. They

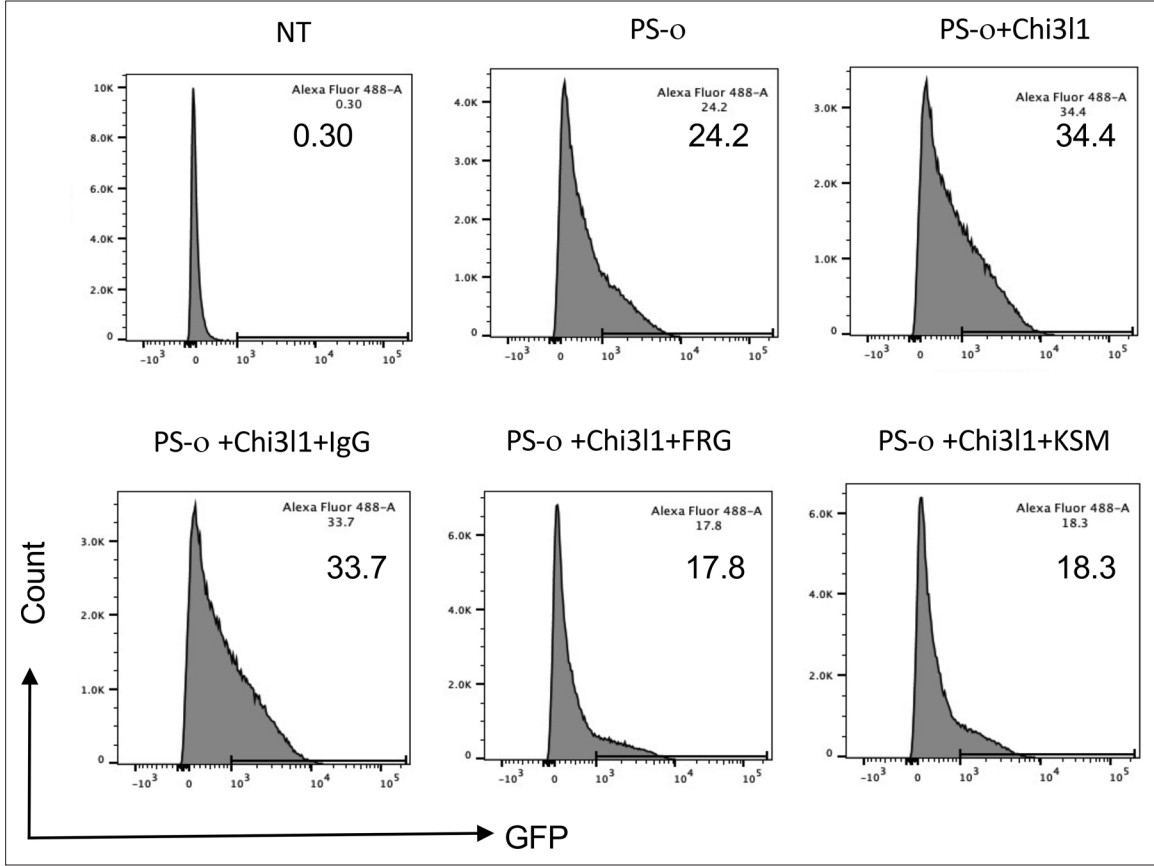

**Figure 7.** Effects of FRG and kasugamycin on omicron pseudovirus infection. Calu-3 cells were incubated with rhCHI3L1 (CHI3L1, 250 ng/ml) or vehicle control for 48 hr in the presence or absence of anti-CHI3L1 (FRG) or its isotype control (IgG) or kasugamycin or vehicle control. Pseudoviruses that contain S proteins with the omicron mutations (PS- ο ) were added, and GFP was quantitated by fluorescence-activated cell sorting (FACS). The percentage of GFP-positive cells was evaluated by flow cytometry. The noted values are representative of a minimum of three similar evaluations.

also demonstrate that antibody-based and small-molecule inhibitors of CHI3L1 inhibit the infection of human epithelial cells by these major SC2 VOC, including delta and omicron. In combination, they suggest that CHI3L1 is a potential therapeutic target that can be manipulated to prevent or alter the natural history of SC2 infection caused by the current and possible future viral variants that utilize ACE2 and SPP.

The S glycoprotein of SC2 is located on the outer surface of the virion and undergoes cleavage into S1 and S2 subunits. The S1 subunit is further divided into an RBD and an N-terminal domain (NTD), which serve as potential targets for neutralization in response to antisera and/or antibodies induced by vaccines (*Aleem et al., 2021*; *Song et al., 2018*). Genetic variation in SC2 can have important implications for disease pathogenesis, especially if the alterations involve the RBD. In keeping with this concept, the SC2 VOC have impressive mutations in the viral S proteins with alterations in RBD and NTD (*Ou et al., 2021*; *Shen et al., 2021*). Three of the VOC have N501Y alterations, which augment viral attachment to ACE2 and subsequent host cell infection (*Aleem et al., 2021*). Omicron has 37 amino acid mutations in its S protein (*Cameroni et al., 2022*). 15 of these mutations are in the RBD and 9 are in the RBM, which is the subdomain of the RBD that directly interacts with ACE2 (*Cameroni et al., 2022*). When viewed in combination, these studies highlight the importance of the viral S proteins and host ACE2 and SPP in the responses induced by SC2 variants. Because our data demonstrate that CHI3L1 is a potent stimulator of ACE2 and SPP, they also provide a mechanism by which CHI3L1-based interventions can be effective therapies in all SC2 variants that utilize ACE2 and SPP to mediate viral infection.

Antibodies against the SC2 Spike proteins are an evolving and important part of the immune response to SC2 and treatment tool kit against COVID-19. Because the S protein of omicron is heavily mutated, the therapeutic efficacy of vaccine-induced antibodies and commercial monoclonal anti-Spike protein antibodies has been characterized. These studies demonstrated that vaccine-induced antibodies can manifest diminished therapeutic efficacy compared to ancestral and other SC2 variants like delta (*Liu et al., 2022*; *VanBlargan et al., 2021*). In keeping with these findings, antibodies from Regeneron Inc and Eli Lilly Inc have been noted to manifest diminished potency against omicron while manifesting impressive efficacy against the delta and other variants (*Liu et al., 2022*; *VanBlargan et al., 2021*). Our studies demonstrate that the anti-CHI3L1 antibody FRG and kasugamycin, an inhibitor of CHI3L1, decrease the expression of ACE2 and the ability of the ancestral and the alpha, beta, gamma, delta, and omicron variants to infect epithelial cells. This led us to hypothesize that FRG and kasugamycin could decrease the infection and spread of all SC2 variants that utilize ACE2 and SPP to elicit cell infection. In keeping with our findings, recent studies have demonstrated that omicron infection requires ACE2 and that omicron binds to ACE2 more avidly than the binding of delta to ACE2 (*Golcuk et al., 2021*). This supports our contention that interventions that target CHI3L1 can be effective in the treatment of viruses that utilize ACE to infect epithelial cells.

Variants of interest (VOI) are defined as viral variants with specific genetic markers that may alter the transmissibility and/or susceptibility of the virus to vaccination or therapeutic interventions when compared to ancestral strains (*Aleem et al., 2021*). If the features of the variants are subsequently appreciated to exist, the variant is then reclassified as a VOC. As of June 22, 2021, there were seven VOI, including epsilon, zeta, eta, theta, kappa, and lambda. More recently, epsilon and Mu have been reclassified as a VOC (*Aleem et al., 2021*). In all cases, ACE2 is presumed to be needed for infection by these viral variants. In keeping with this presumption, we believe that CHI3L1 will also regulate VOI infection, replication, and symptom generation by altering ACE2 and SPP. Additional experimentation, however, will be required to formally define the roles of CHI3L1 and effects of CHI3L1 blockade on the effects of these moieties.

Studies from our laboratory and others have demonstrated that CHI3L1 is a critical regulator of inflammation and innate immunity and a stimulator of type 2 immune responses, fibroproliferative repair, and angiogenesis (*Lee et al., 2012*; *Zhou et al., 2018*; *Dela Cruz et al., 2012*; *Zhou et al., 2014*; *Kang et al., 2008*; *Kang et al., 2015*; *He et al., 2013*; *Lee et al., 2009*). These studies also demonstrated that CHI3L1 is increased in the circulation of patients who are older than 60 years of age and patients with a variety of comorbid diseases, including obesity, cardiovascular disease, kidney disease, diabetes, chronic lung disease, and cancer (*Dela Cruz et al., 2012*; *Garnero et al., 2005*; *Hakala et al., 1993*; *Johansen et al., 2000*; *Johansen et al., 1993*; *Kucur et al., 2007*; *Lee, 2009*; *Lee et al., 2011*; *Lee and Elias, 2010*; *Matsuura et al., 2011*; *Nordenbaek et al., 1999*; *Ostergaard*

*et al., 2002*). In keeping with these findings, we focused recent efforts on the development of CHI3L1-based interventions for these disorders. One of the most effective was a monoclonal antibody raised against amino acid 223–234 of human CHI3L1, which is now called FRG. There are a number of reasons to believe that FRG can be an effective therapy in COVID-19. First, as noted by our laboratory (*Kamle et al., 2021*) and in the studies noted above, it is a potent inhibitor of CHI3L1 stimulation of ACE2 and SPP that decreases the infection of epithelial cells by SC2. In addition, CHI3L1 is a potent stimulator of type 2 immune responses and type 2 and type 1 immune responses counterregulate each other. As a consequence, anti-CHI3L1 augments type 1 immune responses that have potent antiviral properties. Anti-CHI3L1 also inhibits the abnormal fibroproliferative repair responses that are seen in pathologic tissue fibrosis such as that seen in lungs from patients with COVID-19 who require prolonged mechanical ventilation. The present studies add to these insights by highlighting the ability of FRG to inhibit the infection of epithelial cells by the alpha, beta, gamma, delta, and omicron SC2 VOC. When viewed in combination, these studies suggest that FRG is a potent therapeutic that can be used to prevent or diminish SC2 infection and/or the COVID-19 disease manifestations induced by SC2 and its major variants while augmenting type 1 antiviral responses and controlling tissue fibrosis.

REGEN-COV-2 is a combination of the monoclonal antibodies casirivimab and imdevimab that bind to noncompeting epitopes of the RBD of the S protein of SC2 (*O'Brien et al., 2021*). When administered via a subcutaneous route, REGEN-COV2 markedly decreases the risk of hospitalization or death among high-risk persons with COVID-19 (*O'Brien et al., 2021*). Subcutaneous REGEN-COV2 also prevents symptomatic infection in previously uninfected household contracts of infected persons and decreases the duration of the symptoms and the titers of the virus after SC2 infection (*O'Brien et al., 2021*). Because the SC2 VOC have S protein mutations that involve the RBD, one can appreciate the importance of combining two antibodies that target different RBD epitopes to allow these antibodies to neutralize the various VOC, including alpha, beta, gamma, delta, and epsilon (*O'Brien et al., 2021*; *Baum et al., 2020*; *Copin et al., 2021*; *Wang et al., 2021*). Because FRG and casirivimab/imdevimab control SC2 via different mechanisms, it is tempting to speculate that additive or synergistic antiviral and/or anti-disease effects, including pre-exposure and post-exposure prophylaxis, will be seen when FRG and REGEN-COV2 are administered simultaneously. One can also see how the administration of FRG and REGEN-COV2 in combination could protect against the selection of resistant SC2 variants (*O'Brien et al., 2021*).

Kasugamycin was discovered in 1965 in *Streptomyces kasugaensis* and has proven to have antibacterial and antifungal properties (*Takeuchi et al., 1965*; *Umezawa et al., 1965*). Since the 1960s, it has been employed as a pesticide to combat agricultural diseases like rice blast fungus and, as a result, has been extensively studied by the Environment Protection Agency (EPA) (*Health Effects Division, 2005*). Most recently, kasugamycin was shown to inhibit influenza and other viral infections (*Gopinath et al., 2018*). Previous studies from our laboratory have added to our understanding of kasugamycin by demonstrating that it is a powerful inhibitor of CHI3l1 induction of ACE2 and SPP that also inhibits type 2 adaptive immune responses and pathological fibrosis (*Kamle et al., 2021*; *Lee et al., 2021*). Importantly, the studies in this submission go further by demonstrating that these CHI3L1-based effects of kasugamycin can be seen in the ancestral and alpha, beta, gamma, delta, and omicron SC2 VOC. When viewed in combination, these observations suggest that kasugamycin can be used as a prophylactic or therapeutic in COVID-19. This is an interesting concept because kasugamycin can be given via an intravenous or oral route and is known to have minimal toxicity in man (*Takeuchi et al., 1965*; *Ujváry, 2010*).

Our studies demonstrate that cellular infection with SC2, in its ancestral and VOC forms, is diminished by anti-CHI3L1 and kasugamycin. This raises the exciting possibility that these approaches and related reagents could be effective therapeutics. Unfortunately, these studies are limited by our lack of in vivo confirmation. This is due, in part, to our lack of access to a BSL 3 lab facility. It is also due, at least in part, to the known differences between murine and human Ace2 that limit the utility of mice as an in vivo model of SC2 infection. To address this limitation, investigators have used the cytokeratin-18 (K18) promoter to generate K18-hACE2 transgenic mice (*Winkler et al., 2020*; *Dong et al., 2022*). This approach has allowed us to further our understanding of the in vivo tissue effects of SC2. However, it does not meet the needs of our studies because the use of the K18 promoter allows us to define the pathways and regulators that control K18 but does not address the regulation of human ACE2 by moieties such as CHI3L1. We look forward to additional investigation that will address these issues.

At the onset of the SC2 pandemic, there was an urgency to mitigate this new viral illness. Since then significant progress has been made in the treatment of COVID-19 due to intense research efforts that resulted in novel therapeutics and vaccine development at an unprecedented rate (*Aleem et al., 2021*). The progress that was made, however, was diminished by the appearance of SC2 viral variants, particularly delta and omicron. It is now known that SC2 infection results in a disease with two phases. The early phase is characterized by cell infection and viral replication, and the latter phase is characterized by a robust host antiviral immune response (*Aleem et al., 2021*). Current therapies that are used in the early phase of SC2 infection include antivirals like remdesivir and anti-SC2 monoclonal antibody pairings like bamlanivimab/etesevimab and casirivimab/imdevimab. When inflammation and a robust immune response have been triggered, anti-inflammatories like dexamethasone and immunomodulators are available. The present studies add to our understanding of the therapies for the early phase of SC2 by demonstrating that the inhibition of ChI3L1 with FRG and/or kasugamycin ameliorates cellular infection induced by the alpha, beta, gamma, delta, and omicron SC2 VOC. If similar efficiency is seen in vivo, this raises the exciting possibility that FRG or kasugamycin, alone or in combination with each other or other SC2 monoclonal antibodies, can have powerful prophylactic effects and/or inhibit viral infection in SC2-exposed individuals. They also demonstrate that FRG and kasugamycin can directly diminish viral replication and, by decreasing viral load, decrease disease pathology and severity. Additional studies of the importance of CHI3L1 and its roles in infections caused by SC2 variants are warranted.

## Materials and methods

### Cell lines and primary cells in culture

Calu-3 (HTB-55) lung epithelial cells were purchased from American Tissue Type Collection (ATCC) and maintained at 37°C in Dulbecco's modified eagle medium supplemented with high glucose, L-glutamine, minimal essential media nonessential amino acids, penicillin/streptomycin, and 10% fetal bovine serum until used. The Calu-3 cells were authenticated through STR profiling and mycoplasma testing by ATCC.

### Generation of monoclonal antibodies against CHI3L1 (FRG)

The murine monoclonal anti-CHI3L1 antibody (FRG) was generated using peptide antigen (amino acid 223–234 of human CHI3L1) as immunogen through Abmart Inc (Berkeley Heights, NJ). This monoclonal antibody specifically detects both human and mouse CHI3L1 with high affinity (kd ≈ $1.1 \times 10^{-9}$). HEK-293T cells were transfected with the FRG construct using Lipofectamine 3000 (Invitrogen, # L3000015). Supernatant was collected for 7 days, and the antibody was purified using a protein A column (Thermo Fisher Scientific, #89960). Ligand-binding affinity and sensitivity were assessed using ELISA techniques.

### Infection of pseudoviruses with S protein mutations

Pseudoviruses with wildtype S proteins or the S mutations that are seen in the alpha, beta, gamma, and delta variants were purchased from BPS Bioscience Inc (San Diego, CA). The pseudovirus with omicron S protein was obtained from eEnzyme (Gaithersburg, MD). Pseudoviruses containing S protein mutations of COVID variants used in this study can be seen in *Supplementary file 1*. These pseudotyped SARS-CoV-2 virus moieties had a lentiviral core expressing green fluorescent protein (GFP) and the SARS-CoV-2 Spike protein but lacked core SC2 sequences. We then compared the ability of pseudotyped virus with mutated S and ancestral S proteins to infect untreated and/or treated Calu-3 epithelial cells. A plasmid expressing VSV-G protein instead of the S protein was used to generate a pantropic control lentivirus. SARS-CoV-2 pseudovirus or VSV-G lentivirus were used to spin-infect Calu-3 cells in a 6-well plate (931 g for 2 hr at 30°C in the presence of 8 µg/ml polybrene). Flow cytometry analysis of GFP (+) cells was carried out 48 hr after infection on a BD LSRII flow cytometer and analyzed with the FlowJo software.

### Immunofluorescence assay (immunocytochemistry)

Immunofluorescent staining was used to assess cellular integration of pseudoviruses associated with expression of ACE2. Briefly, Calu-3 cells were cultured in 4-well chamber slides ($10^6$ cell/well) for 24 hr

then infected with control and pseudoviruses for 48 hr. The cells on the slides were fixed, permeabilized, and treated with blocking buffer, then incubated with anti-ACE2 antibody (R&D, AF933) overnight at 4°C. The images of cellular immunofluorescence of GFP (+) pseudovirus and Cy-5 (+) ACE2 expression were taken with fluorescent microscopes (Nikon, Eclipse Ti).

## Western blotting (immunoblotting)

25 µg lung or cell lysates were subjected to immunoblot analysis using antibodies against phosphorylated (p) ERK (pERK), total ERK(ERK), phosphorylated (p) AKT (pAKT), total AKT (AKT) (Cell Signaling Tech, MA). These samples were gel fractionated, transferred to membranes, and evaluated as described previously by our laboratory (*Lee et al., 2004*).

## Quantification and statistical analysis

Statistical evaluations were undertaken with GraphPad Prism Software. As appropriate, groups were compared with two-tailed Student's *t*-test or with nonparametric Mann−Whitney *U*-test. Values are expressed as mean ± SEM. One-way ANOVA or nonparametric Kruskal−Wallis tests were used for multiple-group comparisons. Statistical significance was defined as a level of $p < 0.05$.

## Acknowledgements

This work was supported by the National Institute of Health (NIH) grants PO1 HL114501(JAE) and R01 HL115813 (CGL) from NHLBI. This work was also supported by COVID-19 Research Seed Grant from Brown University (CGL).

## Additional information

### Competing interests

Jack A Elias: is a cofounder of Elkurt Pharmaceuticals and Ocean Biomedical which develop therapeutics based on the 18 glycosyl hydrolase gene family. The other authors declare that no competing interests exist.

### Funding

| Funder | Grant reference number | Author |
|---|---|---|
| Brown University | Research Seed Grant | Chun Geun Lee |
| Brown University | GR300201 | Chun Geun Lee |
| National Institutes of Health | PO1 HL114501 | Jack A Elias |
| National Institutes of Health | R01 HL115813 | Chun Geun Lee |

The funders had no role in study design, data collection and interpretation, or the decision to submit the work for publication.

### Author contributions

Suchitra Kamle, Conceptualization, Data curation, Formal analysis, Investigation, Methodology, Resources, Writing – review and editing; Bing Ma, Gail Schor, Investigation, Methodology, Resources; Chang Min Lee, Data curation, Methodology, Resources; Yang Zhou, Methodology, Resources; Chun Geun Lee, Jack A Elias, Conceptualization, Funding acquisition, Project administration, Resources, Writing – original draft, Writing – review and editing

### Author ORCIDs

Suchitra Kamle http://orcid.org/0000-0001-9847-1211
Chun Geun Lee http://orcid.org/0000-0002-9514-3658
Jack A Elias http://orcid.org/0000-0002-3124-8557

Decision letter and Author response

Decision letter https://doi.org/10.7554/eLife.78273.sa1

Author response https://doi.org/10.7554/eLife.78273.sa2

## Additional files

### Supplementary files

• Supplementary file 1. Pseudoviruses containing S protein mutations of COVID variants used in this study.

• Transparent reporting form

### Data availability

Figure 3-source data. Immunocytochemical evaluation of delta pseudovirus infection of Calu-3 cells (with FRG Ab Tx). Figure 6-source data. Immunocytotochemical evaluation of delta pseudovirus infection of Calu-3 cells (with Kasugamycin Tx) . Uncut original gel photos of Western blots used in Figures 4A and 4B have been provided as a supporting document.

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
