## [Editor Report]

In this article, Kamle and colleagues report that inhibition of host constitutively expressed chitinase 3-like-1 (CHI3L1) increased epithelial expression of ACE2 and SPP, resulting in epithelial cell viral uptake of pseudoviruses that express the α, β, γ, δ, or omicron S proteins, and they further show that antagonism of CHI3L1 using anti-CHI3L1 or kasugamycin inhibits epithelial cell infection by the pseudoviruses with ancestral, α, β, γ S protein mutations. The in vitro data have relevance to SARS-CoV-2 pathogenesis and potentially has therapeutic implications in that the anti-CHI3L1 antibody and/or kasugamycin might be a treatment for this pandemic virus. These in vitro data are novel, and the results are clear and convincing. The authors acknowledge the limitation of the lack of in vivo data, and the hope is that the publication of this study will encourage a collaboration where those data can be obtained.

---

## [Decision Letter]

**Decision letter after peer review:**

Thank you for submitting your article "Host Chitinase 3-like-1 is a Universal Therapeutic Target for SARS-CoV-2 Viral Variants in COVID 19" for consideration by *eLife*. Your article has been reviewed by 2 peer reviewers, and the evaluation has been overseen by a Reviewing Editor and Paul Noble as the Senior Editor. The reviewers have opted to remain anonymous.

Essential revisions:

It would be helpful to know whether CHI3L1 expression is increased with SARS-CoV-2 infection. Is there a positive feedback mechanism at play in which early infection increases CHI3L1 expression, for which CHI3L1 then upregulates ACE2 and SPP, thus creating a positive feedback loop that amplifies disease pathogenesis?

*Reviewer #1 (Recommendations for the authors):*

It would be helpful to know whether CHI3L1 expression is increased with SARS-CoV-2 infection. Is there a positive feedback mechanism at play in which early infection increases CHI3L1 expression, for which CHI3L1 then upregulates ACE2 and SPP, thus creating a positive feedback loop that amplifies disease pathogenesis?

*Reviewer #2 (Recommendations for the authors):*

1. The work seems preliminary and the data would need to be confirmed with a live virus – preferably on human primary cells or organoids.

2. in vivo work would also increase the overall impact of the work.

---

## [Author Response]

Essential revisions:It would be helpful to know whether CHI3L1 expression is increased with SARS-CoV-2 infection. Is there a positive feedback mechanism at play in which early infection increases CHI3L1 expression, for which CHI3L1 then upregulates ACE2 and SPP, thus creating a positive feedback loop that amplifies disease pathogenesis?

To address this question we compared the levels of circulating CHI3L1 in patients presenting to the emergency department at Rhode Island Hospital in Providence, R.I. USA during the early pandemic. These studies demonstrated that SARS-CoV-2 infection was associated with impressive increases in the levels of circulating CHI3L1 and that the levels of CHI3L1 were higher in patients with more severe disease. These findings have recently been published in Bioarchives and JCI Insight (1, 2). Importantly, the induction of CHI3L1 in COVID 19 and its correlation with adverse outcomes has also just been confirmed in a recent publication from Italy (3). Both publications are compatible with and support the concept that the severe acute disease that is seen in these patients is due, at least in part, to SARS-CoV-2 stimulation of CHI3L1 which, in turn, stimulates ACE2 and SPP creating a positive feedback loop as envisioned by the reviewer. It is not clear, however, if CHI3L1 plays a role in long COVID. The degree to which CHI3L1 plays a role in the pathogenesis of long COVID will need to be addressed in separate studies.

Reviewer #1 (Recommendations for the authors):It would be helpful to know whether CHI3L1 expression is increased with SARS-CoV-2 infection. Is there a positive feedback mechanism at play in which early infection increases CHI3L1 expression, for which CHI3L1 then upregulates ACE2 and SPP, thus creating a positive feedback loop that amplifies disease pathogenesis?

This issue has been addressed above and cited below in references 1-3 from our lab and investigators in Italy.

Reviewer #2 (Recommendations for the authors):1. The work seems preliminary and the data would need to be confirmed with a live virus – preferably on human primary cells or organoids.2. In vivo work would also increase the overall impact of the work.

The issues have been commented on above.

References Cited:

1. Kamle S, Ma B, He CH, Akosman B, Zhou Y, Lee CM, El-Deiry WS, Huntington K, Liang O, Machan J, Kang M-J, Shin HJ, Mizoguchi E, Lee CG, Elias JA. 2021. Chitinase 3-like-1 is a Therapeutic Target That Mediates the Effects of Aging in COVID-19. *bioRxiv*: 2021.01.05.425478

2. Kamle S, Ma B, He CH, Akosman B, Zhou Y, Lee C-M, El-Deiry WS, Huntington K, Liang O, Machan JT, Kang M-J, Shin HJ, Mizoguchi E, Lee CG, Elias JA. 2021. Chitinase 3-like-1 is a therapeutic target that mediates the effects of aging in COVID-19. *JCI insight* 6: e148749

3. De Lorenzo R, Sciorati C, Lorè NI, Capobianco A, Tresoldi C, Cirillo DM, Ciceri F, Rovere-Querini P, Manfredi AA. 2022. Chitinase-3-like protein-1 at hospital admission predicts COVID-19 outcome: a prospective cohort study. In *Scientific reports*, pp. 7606